# The Ca²⁺ concentration impacts the cytokine production of mouse and human lymphoid cells and the polarization of human macrophages *in vitro*

**Yusuf Cem Eskiocak**[1�], **Zeynep Ozge Ayyildiz**[1,2�], **Sinem Gunalp**[1,2‡], **Asli Korkmaz**[1,2‡], **Derya Goksu Helvaci**[3‡], **Yavuz Dogan**[4], **Duygu Sag**[1,2,5], **Gerhard Wingender**[1] *

1 Izmir Biomedicine and Genome Center (IBG), Balcova/Izmir, Turkey, 2 Department of Genome Sciences and Molecular Biotechnology, Izmir International Biomedicine and Genome Institute, Dokuz Eylul University, Balcova/Izmir, Turkey, 3 School of Medicine, Dokuz Eylul University, Balcova/Izmir, Turkey, 4 Department of Microbiology, Faculty of Medicine, Dokuz Eylul University, Balcova/Izmir, Turkey, 5 Department of Medical Biology, Faculty of Medicine, Dokuz Eylul University, Balcova/Izmir, Turkey

☯ These authors contributed equally to this work.
‡ SG, AK and DGH also contributed equally to this work.
* gerhard.wingender@ibg.edu.tr

**Data Availability Statement:** All relevant data are within the paper and its Supporting Information files.

## Abstract

Various aspects of the *in vitro* culture conditions can impact the functional response of immune cells. For example, it was shown that a Ca²⁺ concentration of at least 1.5 mM during *in vitro* stimulation is needed for optimal cytokine production by conventional αβ T cells. Here we extend these findings by showing that also unconventional T cells (invariant Natural Killer T cells, mucosal-associated invariant T cells, γδ T cells), as well as B cells, show an increased cytokine response following *in vitro* stimulation in the presence of elevated Ca²⁺ concentrations. This effect appeared more pronounced with mouse than with human lymphoid cells and did not influence their survival. A similarly increased cytokine response due to elevated Ca²⁺ levels was observed with primary human monocytes. In contrast, primary human monocyte-derived macrophages, either unpolarized (M0) or polarized into M1 or M2 macrophages, displayed increased cell death in the presence of elevated Ca²⁺ concentrations. Furthermore, elevated Ca²⁺ concentrations promoted phenotypic M1 differentiation by increasing M1 markers on M1 and M2 macrophages and decreasing M2 markers on M2 macrophages. However, the cytokine production of macrophages, again in contrast to the lymphoid cells, was unaltered by the Ca²⁺ concentration. In summary, our data demonstrate that the Ca²⁺ concentration during *in vitro* cultures is an important variable to be considered for functional experiments and that elevated Ca²⁺ levels can boost cytokine production by both mouse and human lymphoid cells.

## Introduction

Various cell media have been developed for *in vitro* cell cultures to optimize the growth and survival of particular cell types. For example, the RPMI1640 media is frequently used for *in*

**Funding:** This work was funded by grants from the Scientific and Technological Research Council of Turkey (TUBITAK, #117Z216, GW), the European Molecular Biology Organization (EMBO, #IG3073; GW), and the H2020 Marie Sklodowska-Curie Actions (#777995, GW, DS). The funders had no role in study design, data collection and analysis, decision to publish or preparation of the manuscript.

**Competing interests:** The authors have declared that no competing interests exist.

**Abbreviations:** αGalCer, α-galactosylceramide; FCS, fetal calf serum; ICCS, intracellular cytokine staining; iNKT, invariant Natural Killer T; MAIT, mucosal-associated invariant T; MFI, mean fluorescent intensity; PBMCs, peripheral blood mononucleated cells; ROS, reactive oxygen species; RT, room temperature.

*vitro* cultures of mouse and human lymphocytes [1–3]. However, it was suggested that the Ca$^{2+}$ concentration of RPMI1640 (0.49 mM) is actually suboptimal for the *in vitro* stimulation of conventional mouse [4] and human [5] αβ T cells, as measured by cytokine production, and that a 1 mM CaCl$_2$ supplement is required to obtain the maximal cytokine response. Whether the function of unconventional T cells or of other lymphoid and myeloid cells similarly is impacted by the Ca$^{2+}$ concentration *in vitro* is currently unknown. Unconventional T cells differ from conventional αβ T cells by their development and functional capabilities. Prominent examples of unconventional T cells are invariant Natural Killer T (*i*NKT) cells and mucosal-associated invariant T (MAIT) cells, which both express an αβTCR, and γδ T cells, which express a γδTCR. Both *i*NKT and MAIT cells express a highly conserved invariant TCR α-chain, which recognizes glycolipids or riboflavin derivates in the context of the non-polymorphic MHC class I homologs CD1d or MR1, respectively [6–9]. γδ T cells are largely MHC-unrestricted and although the antigen for many γδ T cells is not known, some respond to phosphorylated isoprenoid metabolites or lipids [10, 11]. These unconventional T cells develop as memory T cells and can provide a first line of defence during immune responses [12]. B cells are the second main adaptive lymphoid cell type and are characterized by the expression of a BCR [13]. As an example of myeloid cells, we choose here macrophages, which are phagocytic and antigen-presenting effector cells of the innate immune system [14]. Depending on the way of stimulation, macrophages can differentiate into several functionally distinct subsets, often referred to as classically activated M1 or alternatively activated M2 macrophages [14–16]. To determine the impact of the Ca$^{2+}$ concentration on lymphoid and myeloid cells besides conventional αβ T cells, we here compared their immune response *in vitro* in the presence of normal RPMI1640 medium (RPMI$^{norm}$) and RPMI1640 medium supplemented with 1 mM Ca$^{2+}$ (RPMI$^{suppl}$). Our data indicated that elevated Ca$^{2+}$ concentrations during PMA/ionomycin stimulation *in vitro* increased the cytokine production by both mice and human lymphoid cells for most cytokines tested. Furthermore, the polarization of human macrophages shifted towards an M1 phenotype in the high-Ca$^{2+}$ environment. Consequently, the Ca$^{2+}$ concentration during *in vitro* cultures is an important variable to be considered for functional experiments.

## Results

### Cytokine production of mouse unconventional T cells and of B cells is augmented by increased Ca$^{2+}$ concentrations *in vitro*

Following activation, *i*NKT cells are able to produce a wide range of cytokines, including T$_h$1 cytokines, like IFNγ and TNF; T$_h$2 cytokines, like IL-4 and IL-13, the T$_h$17 cytokine IL-17A, as well as IL-10 [17–19]. When splenic mouse *i*NKT cells (**S1 Fig**) were stimulated *in vitro* with PMA and ionomycin, the increased Ca$^{2+}$ concentration had no detrimental effect on *i*NKT cell survival (**S2A Fig**). However, we observed a marked and significant increase in the production of all cytokines tested (IFNγ, IL-2, IL-4, IL-10, IL-13, IL-17A) by *i*NKT cells when stimulated in media supplemented with calcium (**Fig 1**). Similar to *i*NKT cells, γδ T cells can produce a wide range of cytokines following stimulation [20]. No detrimental effect of the Ca$^{2+}$ supplementation on γδ T cell survival was observed (**S2B Fig**). However, we noticed a significant increase in the production of IFNγ, IL-2, and IL-4 by γδ T cells stimulated in elevated Ca$^{2+}$ levels (**Fig 2A–2C**). In contrast, the changes for IL-10 and IL-17A remained non-significant (**S3A and S3B Fig**). We also analysed the impact of Ca$^{2+}$ levels on B cells. The survival of stimulated B cells was not impaired by the Ca$^{2+}$ concentration *in vitro* (**S2C Fig**). However, an increase in the production of IL-2 and IL-10 was observed (**Fig 2D and 2E**), while IFNγ (**S3C Fig**) remained unaffected. Therefore, the PMA/ionomycin stimulation of mouse lymphoid

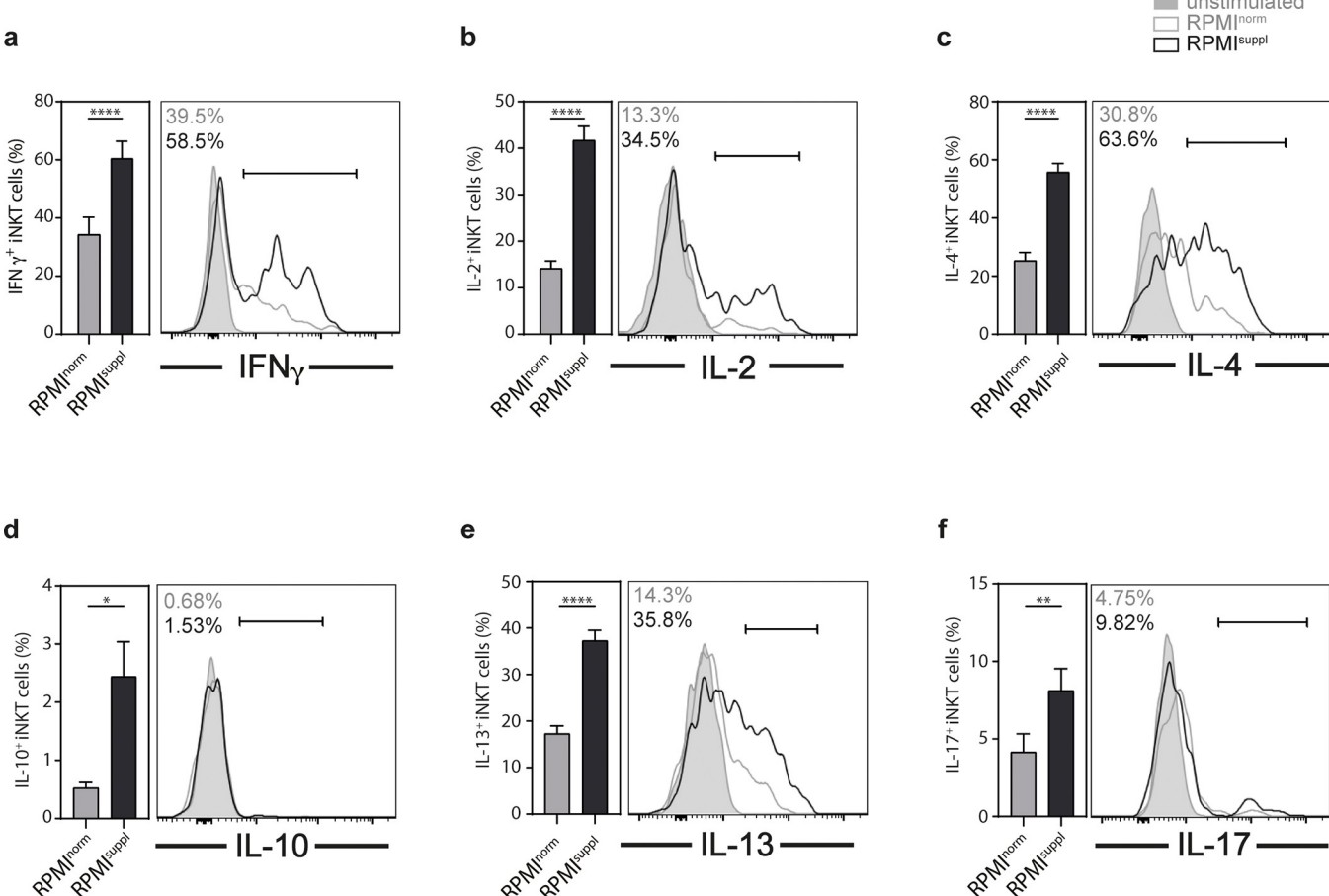

**Fig 1. Ca²⁺ supplementation *in vitro* increases the cytokines production of mouse *i*NKT cells.** Splenocytes from C57BL/6 mice were stimulated 4 h with 50 ng/ml PMA and 1 µg/ml ionomycin in either normal RPMI1640 medium (RPMI^norm) or RPMI1640 medium supplemented with 1 mM Ca²⁺ (RPMI^suppl). The production of (**a**) IFNγ, (**b**) IL-2, (**c**) IL-4, (**d**) IL-10, (**e**) IL-13, and (**f**) IL-17A by *i*NKT cells (live CD8α⁻ CD19/CD45R⁻ CD44⁺ TCRβ/CD3ε⁺ CD1d/PBS57-tetramer⁺ cells) was analysed by intracellular cytokine staining (ICCS). Summary graphs (left panels) and representative data (right panels) from gated *i*NKT cells are shown, respectively. Data were pooled from three independent experiments with three mice per group per experiment (n = 9).

cells in complete RPMI medium supplemented with 1.0 mM Ca²⁺ improves the detection of numerous cytokines, without impacting the survival of the cells.

## Cytokine production of human unconventional T cells and of B cells is augmented by increased Ca²⁺ concentrations *in vitro*

Having established that increased Ca²⁺ concentrations during *in vitro* stimulation can augment the cytokine production of mouse lymphoid cells, we next tested its effect on human lymphoid cells. For primary human *i*NKT cells, the elevated Ca²⁺ levels only increased the production of TNF slightly (**Fig 3A**), without effects on the other cytokines tested (IL-2, IFNγ, IL-4, IL-17A) or on the expression of the activation marker CD69 (**S4 Fig**). For primary human Vδ2⁺ T cells, some (TNF, IFNγ) (**Fig 3B and 3c**) but not all (IL-2, IL-4) were boosted by the Ca²⁺ supplementation, which did also not influence CD69 expression (**S5A–S5C Fig**). For primary human MAIT cells, the elevated Ca²⁺ levels increased the production of all cytokines tested (IL-2, TNF, IFNγ) (**Fig 3D–3F**), without changes to the expression of CD69 (**S5D Fig**). A similar effect of the Ca²⁺ concentrations was noted for primary human B cells (IL-2, TNF, CD69) (**Fig 4**, **S5E Fig**). For some immune cells, the low frequency in the peripheral blood

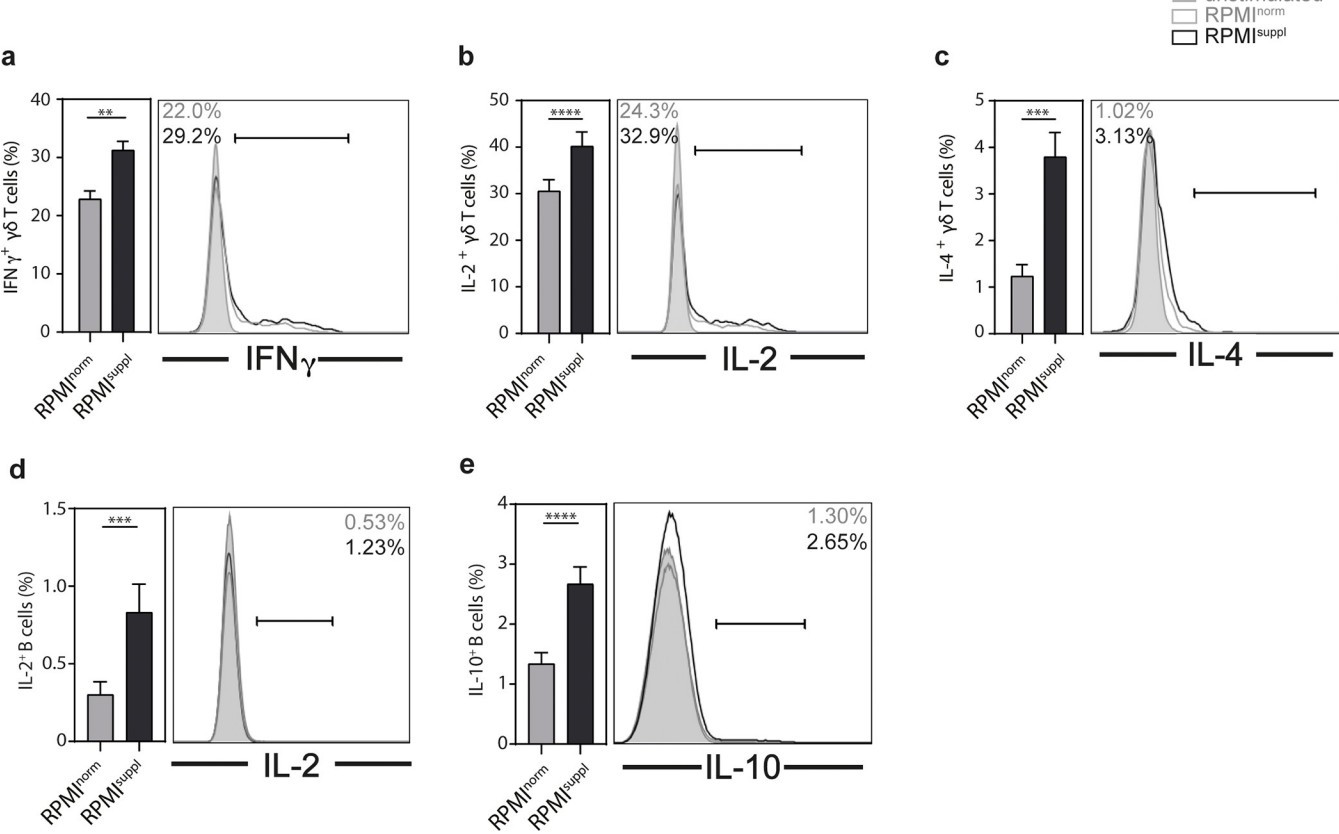

**Fig 2. Ca²⁺ supplementation *in vitro* increases the production of some cytokines by mouse γδ T and B cells.** Splenocytes from C57BL/6 mice were stimulated 4 h with 50 ng/ml PMA and 1 μg/ml ionomycin in either normal RPMI1640 medium (RPMI^norm) or RPMI1640 medium supplemented with 1 mM Ca²⁺ (RPMI^suppl). The production of **(a)** IFNγ, **(b)** IL-2, and **(c)** IL-4 by γδ T cells (live CD19/CD45R⁻ CD4⁻ CD8α⁻ CD3ε⁺ γδTCR⁺ cells) and the production of **(d)** IL-2 and **(e)** IL-10 by B cells (live CD3ε⁻ CD4⁻ CD8α⁻ CD19/CD45R⁺ cells) was analysed by ICCS. Summary graphs (left panels) and representative data (right panels) from gated γδ T cells and B cells are shown, respectively. Data were pooled from three independent experiments with three mice per group per experiment (n = 9).

makes their analysis directly *ex vivo* difficult, which is why protocols were established to expand them *in vitro*. We, therefore, also tested the impact of elevated Ca²⁺ concentrations on *in vitro* expanded *i*NKT and Vδ2⁺ T cells. For expanded human *i*NKT cells, the elevated Ca²⁺ levels only increased the production of TNF slightly (**Fig 5B**), without effects on the other cytokines tested (IL-2, IFNγ, IL-4, IL-17A) (**S6A–S6D Fig**) and decreased CD69 expression (**Fig 5A**). For expanded human Vδ2⁺ T cells, some (IFNγ, GM-CSF) (**Fig 5D and 5E**) but not all (CD69, TNF, IL-4) (**S6E–S6G Fig**) markers were boosted by the Ca²⁺ supplementation, whereas the production of IL-2 surprisingly decreased (**Fig 5C**). For all human lymphoid cell populations tested, the Ca²⁺ supplementation *in vitro* did not impair the cell survival (**S7A–S7F Fig**). Therefore, similar to mouse lymphoid cells, the detection of cytokines in primary human lymphoid cells can be improved by increasing the Ca²⁺ levels during the *in vitro* stimulation.

## Increased Ca²⁺ during *in vitro* stimulation of macrophages increases cell death and induces a shift toward M1 polarization

Given the clear ability of increased *in vitro* Ca²⁺ concentrations to augment the cytokine production of stimulated mouse and human lymphoid cells, we tested next the impact on myeloid cells. To this end, we initially measured the impact of elevated Ca²⁺ levels on primary human

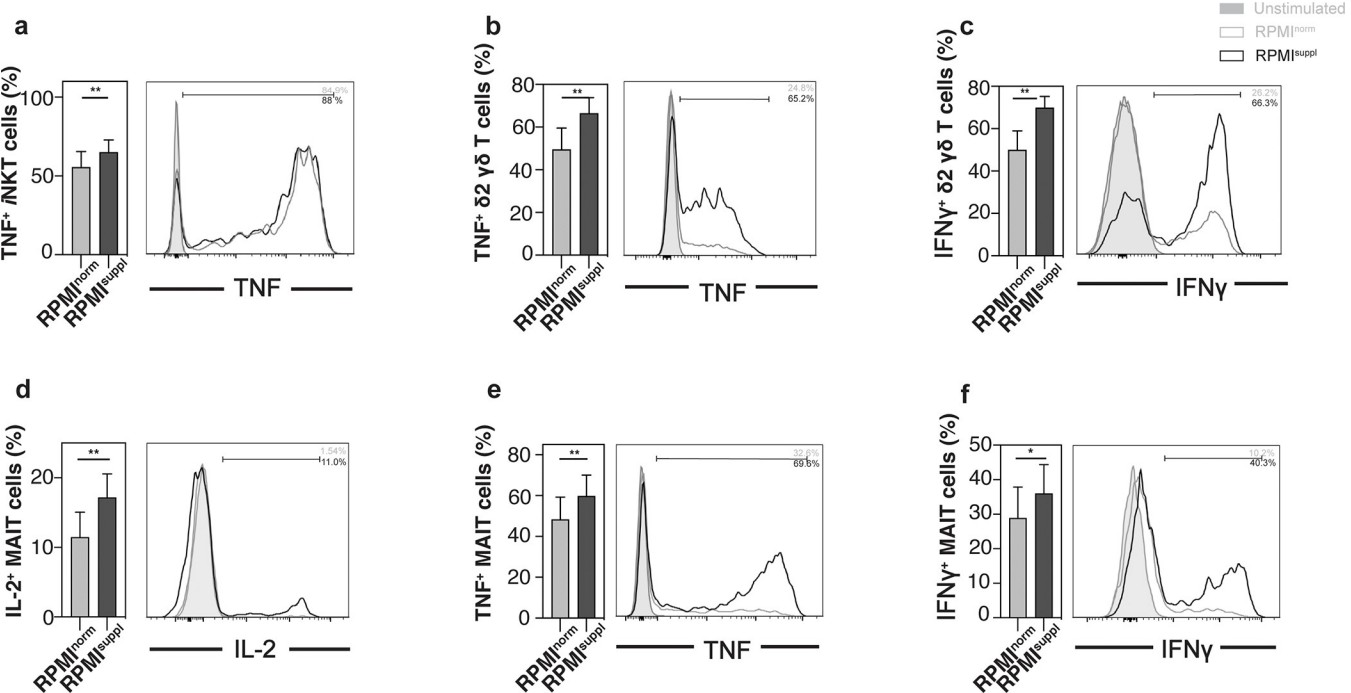

**Fig 3. Ca²⁺ supplementation *in vitro* modulates cytokine production by primary human *i*NKT cells, Vδ2⁺ T cells, and MAIT cells.** PBMCs were isolated from the residual leukocyte units of healthy donors. PBMCs were stimulated for 4 h with 25 ng/ml PMA and 1 μg/ml ionomycin in either normal RPMI1640 medium (RPMInorm) or RPMI1640 medium supplemented with 1 mM Ca²⁺ (RPMIsuppl). **(a)** Human Vα24*i* NKT cells (live CD14⁻ CD20⁻ CD3⁺ 6B11⁺ cells) were analysed for the expression of TNF. **(b, c)** Human Vδ2⁺ T cells (live CD14⁻ CD20⁻ CD3⁺ γδTCRlow or Vδ2⁺ cells) were analysed for the expression of (b) TNF and (c) IFNγ. **(d-f)** Human MAIT cells (live CD14⁻ CD20⁻ CD3⁺ Vα7.2⁺ CD161⁺ cells) were analysed for the production of (d) IL-2, (e) TNF, and (f) IFNγ. Summary graphs (left panels) and representative data (right panels) from gated *i*NKT cells, Vδ2⁺ T cells, and MAIT cells are shown, respectively. Data were pooled from three (*i*NKT cells, Vδ2⁺ T cells; n = 9) and four independent (MAIT cells; n = 12) experiments with three samples each.

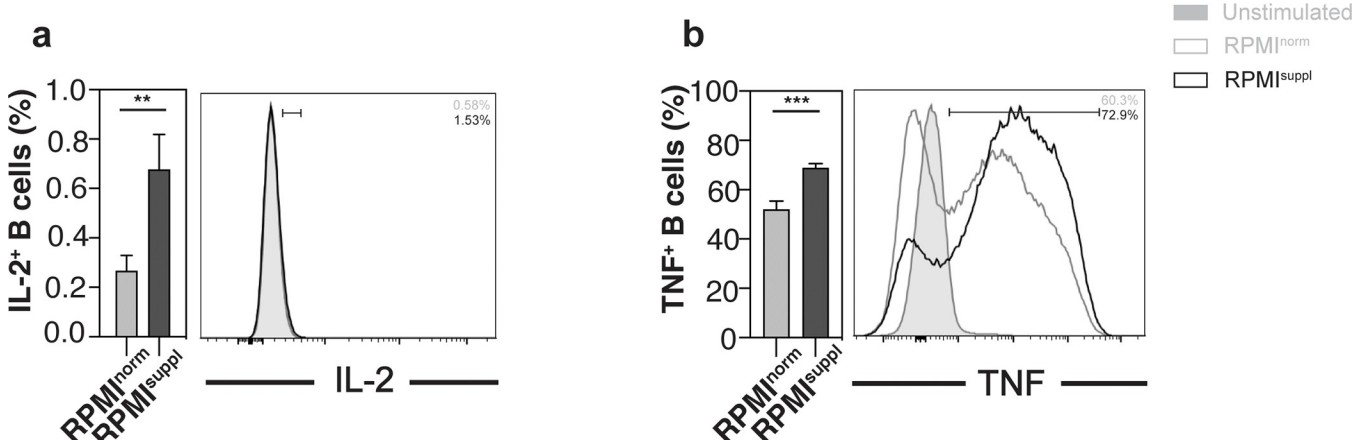

**Fig 4. Ca²⁺ supplementation *in vitro* increases the cytokine production of primary human B cells.** PBMCs were isolated from residual leukocyte units of healthy donors. PBMCs were stimulated for 4 h with 25 ng/ml PMA and 1 μg/ml ionomycin in either normal RPMI1640 medium (RPMInorm) or RPMI1640 medium supplemented with 1 mM Ca²⁺ (RPMIsuppl). Human B cells (CD14⁻ CD3⁻ CD20⁺ cells) were analysed for the production of **(a)** IL-2 and **(b)** TNF. Summary graphs (left panels) and representative data (right panels) from gated B cells are shown, respectively. Data were pooled from four independent experiments with three samples each (n = 12).

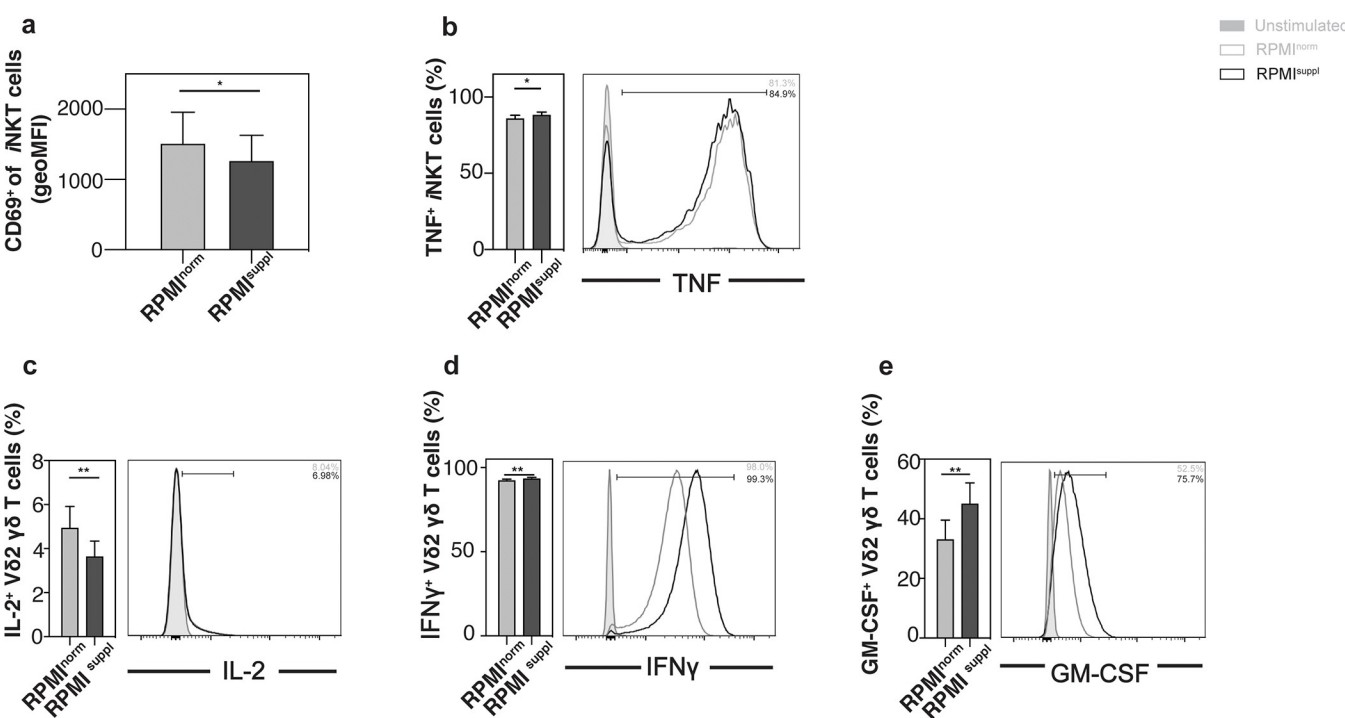

**Fig 5. Ca²⁺ supplementation *in vitro* modulates cytokine production by expanded human *i*NKT cells and Vδ2⁺ T cells. (a, b)** *i*NKT cells were expanded *ex vivo* in the presence of αGalCer. The expanded cells were stimulated for 4 h with 25 ng/ml PMA and 1 µg/ml ionomycin in either normal RPMI1640 medium (RPMI^norm) or RPMI1640 medium supplemented with 1 mM Ca²⁺ (RPMI^suppl). Vα24*i* NKT cells (live CD14⁻ CD20⁻ CD3⁺ 6B11⁺ cells) were analysed for the expression of (a) CD69 and the production of (b) TNF. **(c-e)** Vδ2⁺ T cells were expanded *in vitro* in the presence of Zoledronic acid. Human Vδ2⁺ T cells (live CD14⁻ CD20⁻ CD3⁺ Vδ2⁺ cells) were analysed for the production of (c) IL-2, (d) IFNγ, and (e) GM-CSF. Summary graphs (left panels) and representative data (right panels) from gated *i*NKT cells and Vδ2⁺ T cells are shown, respectively. Data were pooled from four (*i*NKT cells; n = 12) and three (Vδ2⁺ T cells, n = 9) independent experiments with three samples each.

PBMC monocytes and noticed that some (TNF, IFNγ; **Fig 6**) but not all (IL-2; **S5F Fig**) cytokines were boosted by the Ca²⁺ supplementation, without impact on cell survival (**S7G Fig**). These data suggested that the cytokine detection by primary myeloid cells could benefit from *in vitro* Ca²⁺ supplementation as well. Blood monocytes have the ability to differentiate into functionally distinct macrophages subsets and, therefore, we tested next the impact of the Ca²⁺ concentration on human monocyte-derived macrophages. Primary human monocyte-derived macrophages (M0 macrophages) were polarized into M1 with LPS and IFNγ, into M2a with IL-4 stimulation, or into M2c with IL-10 with or without Ca²⁺ supplementation. Surprisingly, and in contrast to the findings with human lymphoid cells (**S7 Fig**), the polarization of M0 macrophages in elevated Ca²⁺ concentrations increased cell death, regardless of the subtype they were polarized into (**Fig 7A–7c**). Similar results were seen when M0 macrophages were cultured alone in Ca²⁺ supplemented medium (**S8 Fig**). When M0 macrophages were cultured alone or when they were polarized into M1 macrophages, higher Ca²⁺ levels increased the expression of the M1 markers HLA-DRα and CD86 (**S9A Fig**). In contrast, when M0 macrophages were cultured alone or when they were polarized into M2a macrophages, Ca²⁺ supplementation decreased the expression of the M2a markers CD200R and CD206 (**S9B Fig**). The Ca²⁺ levels did not influence the expression of CD163 on M2c polarized macrophages (**S9C Fig**). Importantly, when the expression of the M1 markers HLA-DRα and CD86 was analysed on M2a and M2c macrophages, Ca²⁺ supplementation increased the expression of HLA-DRα on M2a macrophages (**Fig 7D**) and of CD86 on both M2 macrophages subsets (**Fig 7E and 7F**). These data indicate that increased Ca²⁺ concentrations support phenotypic M1

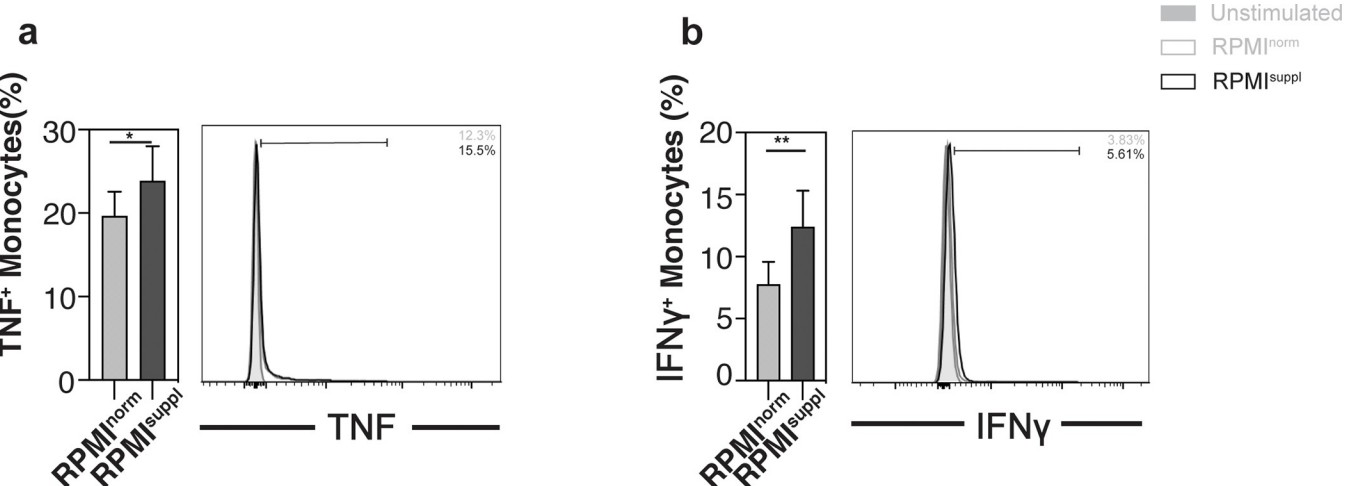

**Fig 6. Ca$^{2+}$ supplementation *in vitro* increases the cytokine production of primary monocytes.** PBMCs were isolated from residual leukocyte units of healthy donors. PBMCs were stimulated for 4 h with 25 ng/ml PMA and 1 µg/ml ionomycin in either normal RPMI1640 medium (RPMI$^{norm}$) or RPMI1640 medium supplemented with 1 mM Ca$^{2+}$ (RPMI$^{suppl}$). Human monocytes (CD3$^-$ CD20$^-$ CD14$^+$ cells) were analysed for production of **(a)** TNF and **(b)** IFNγ. Summary graphs (left panels) and representative data (right panels) from gated monocytes are shown, respectively. Data were pooled from four independent experiments with three samples each (n = 12).

polarization of human monocyte-derived macrophages *in vitro*. However, these phenotypic changes appeared not to translate into functional changes, as the production of TNF and CXCL10 by M1 macrophages (**Fig 7G and 7H**) and the production of TGFβ and IL-4 by M2a and M2c macrophages (**Fig 7I and 7J**) was not influenced by the Ca$^{2+}$ supplementation. These data suggest that for myeloid cells the impact of an increased Ca$^{2+}$ concentration on the cytokine production depends on the activation status of the cells and needs to be tested cell type specifically.

## Discussion

Here we demonstrate that supplementing the RPMI1640 medium with 1 mM Ca$^{2+}$ can increase the cytokine production by both mice and human lymphoid cells during PMA/ionomycin stimulation *in vitro* without impacting the survival of the cells. This effect appeared stronger for primary lymphoid cells than *in vitro* expanded cells. Primary human monocytes responded similarly with an augmented cytokine production to the elevated Ca$^{2+}$ concentrations. However, for human monocyte-derived macrophages a distinct effect was observed: elevated Ca$^{2+}$ concentrations *in vitro* led to increased cell death and promoted phenotypic M1 polarization without impacting cytokine production.

Previous work on conventional mouse [4] and human [21] αβ T cells showed that the 0.49 mM Ca$^{2+}$ in RPMI1640 are suboptimal to drive cytokine production during *in vitro* stimulation and a minimal concentration of 1.5 mM was suggested. The basal level of intracellular Ca$^{2+}$ in T cells is approx. 100 nM and can increases to 1 µM following stimulation [22]. Therefore, it is not self-evident how an increase of the extracellular Ca$^{2+}$ concentration above 0.49 mM can boost cytokine production by conventional αβ T cells [4, 21]. Irrespective of the mechanism, this observation is important to optimize the analysis of conventional αβ T cells *in vitro*. Here, we confirmed that unconventional αβ T cells, γδ T cells, and B cells from both mouse and human also produce more cytokines after *in vitro* stimulation in the presence of elevated Ca$^{2+}$ levels. The Ca$^{2+}$ concentration of the extracellular fluid *in vivo* in humans was measured to be 2.2–2.7 mM [23]. This indicates that the Ca$^{2+}$ supplementation brings the Ca$^{2+}$

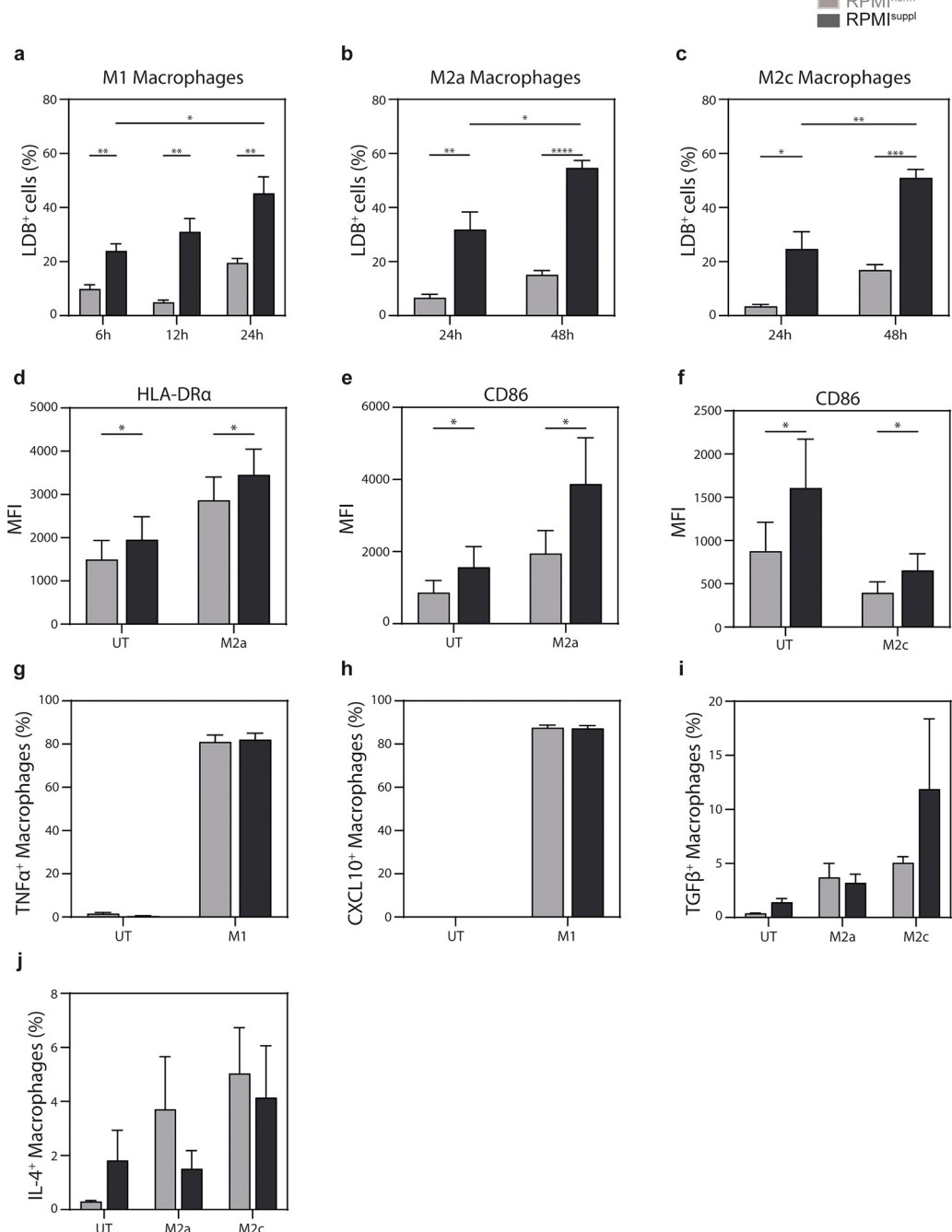

**Fig 7. Ca$^{2+}$ supplementation *in vitro* increases cell death in polarized human macrophages and favours an M1 phenotype.** Primary human monocyte-derived macrophages were cultured untreated (UT) or polarized into M1 (100 ng/ml LPS, 20 ng/ml IFNγ), M2a (20 ng/ml IL-4), or M2c (20 ng/ml IL-10) macrophages for the indicated hours in either normal RPMI1640 medium (RPMI$^{norm}$) or RPMI1640 medium supplemented with 1 mM Ca$^{2+}$ (RPMI$^{suppl}$) as indicated. **(a-c)** The bar graphs show the relative percentages of cells positive for LIVE/DEAD Fixable Blue Dead Cell Stain (LDB$^+$ cells), indicating dead cells, for (a) M1, (b) M2a, and (c) M2c macrophage. **(d-f)** The expression (mean fluorescent intensity, MFI) of (d) HLA-DRα (UT; M2a, 24 hours) and (e, f) CD86 ((e): UT; M2a, 24 hours; (f): UT; M2c, 24 hours) on live macrophages is shown. The data shown are means ± SEM of biological replicates of six donors, pooled from two independent experiments with similar results. **(g-j)** The production of (g) TNF and (h) CXCL10 by M1 macrophages (6 hours) and of (i) TGFβ, and (j) IL4 by M2a (48 hours) or M2c (48 hours) macrophages was analysed ICCS. The data shown are means ± SEM of biological replicates of five (TGFβ, IL-4) or six donors (CXCL10, TNF), pooled from two independent experiments with similar results.

concentration of the RPMI$^{suppl}$ medium (approx. 1.8 mM) close to the physiological conditions *in vivo*.

For primary mouse lymphoid cells, the elevated Ca$^{2+}$ concentration increased the production of 6 out of 6 cytokines tested for *i*NKT cells (Fig 1), 3 out of 5 for γδ T cells (Fig 2A–2C; S3A and S3B Fig), and 2 out of 3 for B cells (Fig 2D and 2E; S3C Fig), without a decrease of the production of any of the cytokines tested. Interestingly, the boosting Ca$^{2+}$ effect appeared weaker for primary human lymphoid cells: the elevated Ca$^{2+}$ concentration increased the production of 1 out of 5 cytokines tested for *i*NKT cells (Fig 3A; S4B–S4E Fig), 2 out of 4 for γδ T cells (Fig 3B and 3C; S5A–S5C Fig), 3 out of 3 for MAIT cells (Fig 3D–3F), and 2 out of 2 for B cells (Fig 4), again without any decreases of any of the cytokines tested. The reason for this species difference is unclear at this point. The cytokine response of the *in vitro* expanded cell lines under increased Ca$^{2+}$ concentrations was comparable to the primary cells: both primary and expanded *i*NKT cells showed increased production for 1 (TNF) out of 5 cytokines tested (Figs 3A and 5B; S6A–S6D Fig). Primary γδ T cells showed increased production for 2 (IFNγ, TNF) out of 4 (Fig 3B and 3C; S5B and S5C Fig) and expanded γδ T cells showed increased production for 2 (GM-CSF, IFNγ) out of 5 (Fig 5D and 5E; S6F and S6G Fig) cytokines tested. Surprisingly, expanded γδ T cells showed a decrease of IL-2 with increased Ca$^{2+}$ (Fig 5C), which is the only instance in which we noticed a decrease in cytokine production.

Similar to the lymphoid cells, primary human monocytes increased the production of 2 out of 3 cytokines tested (Fig 6; S5F Fig) when stimulated in the presence of elevated Ca$^{2+}$ levels (Fig 6; S5F Fig). However, the data we obtained with human primary monocyte-derived macrophages were in contrast to the ones from lymphoid cells and primary human monocytes. Most importantly, we noticed a clear increase in the frequency of macrophages that died when incubated for more than six hours in the presence of elevated Ca$^{2+}$ levels (Fig 7A–7C; S8 Fig). Calcium-induced ER-stress [24, 25] and mitochondrial changes [26] can trigger ROS- (reactive oxygen species) production in macrophages, which can lead to cell death [24, 25, 27–29]. This might explain why macrophages are more sensitive to the Ca$^{2+}$ concentration of the medium. Furthermore, we observed a shift towards an M1 phenotype for both M0 macrophages (S9 Fig) as well as M1 and M2 macrophages (Fig 7D–7F) in the presence of elevated Ca$^{2+}$ levels. However, these phenotypic changes appeared not to translate into functional changes (Fig 7G–7J). Incidentally, ROS-induced NFκB/MAPK activation in macrophages [30, 31] supports M1 polarization [32–34]. However, given the large amount of macrophage cell death we observed, we cannot exclude the possibility that the apparent M1 shift is the result of M1 macrophages potentially being less sensitive to this Ca$^{2+}$-induced cell death. We are aware of only two other studies on the impact of Ca$^{2+}$ on the cytokine production of macrophages. Both showed that calcium influx can impair LPS-induced IL-12 production of mouse macrophages [35, 36] without affecting the production of TNF or IL-6 [36]. Therefore, it is unclear at this stage whether elevated Ca$^{2+}$ concentration *in vitro* can influence the cytokine production of macrophages.

In summary, our data demonstrate that the Ca$^{2+}$ concentration during *in vitro* cultures is an important variable to be considered for functional experiments and that supplementing the media with 1 mM Ca$^{2+}$ can boost the cytokine production by both mice and human lymphoid cells.

## Material and methods

### Human samples

Residual leukocyte units from healthy donors were provided by Dokuz Eylul University Blood Bank (Izmir, Turkey) after obtaining informed written consent from all donors. The ethical

approval for the study was obtained from the 'Noninvasive Research Ethics Committee' of the Dokuz Eylul University (approval number: 2018/06-27/3801-GOA). All protocols performed were in accordance with the relevant guidelines and regulations for human samples.

## Mice

All mice were housed in the vivarium of the Izmir Biomedicine and Genome Center (IBG, Izmir, Turkey) in accordance with the respective institutional animal care committee guidelines. C57BL/6 and BALB/c mice were originally purchased from the Jackson Laboratories (Bar Harbor, ME, USA). All mouse experiments were performed with prior approval by the institutional ethic committee ('Ethical Committee on Animal Experimentation' of the Izmir Biomedicine and Genome Center, approval number: 19/2016) in accordance with national laws and policies. All the methods were carried out in accordance with the approved guidelines and regulations and following 3R (replacement, reduction, refinement) procedures. These aspects of this study are reported in accordance with the ARRIVE guidelines [37]. Mice were sacrificed by cervical dislocation and as all mouse experiments were performed *ex vivo*, no anaesthesia was required.

## Reagents, monoclonal antibodies, and flow cytometry

α-galactosylceramide (αGalCer) was obtained from Avanti Polar Lipids (Birmingham, AL, USA). Monoclonal antibodies were purchased from either BioLegend (San Diego, CA, USA), eBiosciences (San Diego, CA, USA), BD Biosciences (Franklin Lane, NJ, USA), or R&D Systems (Minneapolis, MN, USA). The list of antibodies against the mouse and human antigens used in this study, with clone name, vendor, and the conjugated fluorochrome, is given in **S1 Table**. Anti-mouse CD16/32 (2.4G2) antibody (Tonbo Biosciences, San Diego, CA, USA) or Human TruStain FcX (BioLegend) was used to block Fc receptors according to the manufacturers' recommendations. Unconjugated mouse and rat IgG antibodies were purchased from Jackson ImmunoResearch (West Grove, PA, USA). Dead cells were labelled with Zombie UV Dead Cell Staining kit (BioLegend) or with LIVE/DEAD Fixable Blue Dead Cell Stain kit (ThermoFisher Scientific, Waltham, MA, USA). Flow cytometry of fluorochrome-conjugated antigen-loaded CD1d tetramers were performed as described [38]. Cells were analysed with LSR-Fortessa (BD Biosciences), and data were processed with CellQuest Pro (BD Biosciences) or FlowJo (BD Biosciences) software. Graphs derived from digital data are displayed using a 'bi-exponential display'. Cell were gated as follows: (a) mouse: Vα14*i* NKT cells (live CD8α$^-$ CD19/CD45R$^-$ CD44$^+$ TCRβ/CD3ε$^+$ CD1d/PBS57-tetramer$^+$ cells), γδ T cells (live CD19/ CD45R$^-$ CD4$^-$ CD8α$^-$ CD3ε$^+$ γδTCR$^+$ cells), and B cells (live CD3ε$^-$ CD4$^-$ CD8α$^-$ CD19/ CD45R$^+$ cells); (b) human: Vα24*i* NKT cells (live CD14$^-$ CD20$^-$ CD3$^+$ 6B11$^+$), δ2$^+$ T cells *ex vivo* (live CD14$^-$ CD20$^-$ CD3$^{high}$ γδTCR$^{low}$ cells, or live CD14$^-$ CD20$^-$ CD3$^+$ Vδ2$^+$ cells) and *in vitro* (live CD14$^-$ CD20$^-$ CD3$^+$ Vδ2$^+$ cells), MAIT cells (live CD14$^-$ CD20$^-$ CD3$^+$ Vα7.2$^+$ CD161$^+$ cells), B cells (live CD3$^-$ CD14$^-$ CD20$^+$ cells), and macrophages (live CD68$^+$ cells). Representative data are shown in **S1 Fig**.

## Cell preparation

Single-cell suspensions from mouse spleens were prepared as described [39]. In brief, the spleens were filtered through a 70 μm cell strainer (BD Biosciences) and red blood cells and dead cells were eliminated through Lymphoprep (StemCell Technologies, Vancouver, Canada) density gradient centrifugation (300 *g*, 10 min, RT). Human PMBCs were obtained from the residual leukocyte units of healthy donors via density gradient centrifugation (300 *g*, 30 min, RT) with Ficoll-Paque Plus (GE Healthcare, Chicago IL, USA). To obtain monocytes, a second

density gradient centrifugation with 46% iso-osmotic Percoll (GE Healthcare) was performed (400 *g*, 10 min, RT) [40].

### *In vitro* expansion of human *i*NKT cells

Human *i*NKT cells were expanded from resting PBMCs of healthy donors as described before [41]. Briefly, freshly isolated PBMCs (1 x 10$^6$ cell/ml, 5 ml/well) were treated with 100 ng/ml αGalCer (KRN7000, Avanti Polar Lipids) and cultured for 13 days. 20 IU/ml human recombinant IL-2 (Proleukin, Novartis, Basel, Switzerland) was added to the cultures every other day starting from day 2. From day 6 onwards, the concentration of IL-2 was increased to 40 IU/ml. At the end of the expansion, an aliquot of each sample was collected and analysed for *i*NKT cell expansion by flow cytometry. Expansion was done in RPMI 1640 (Gibco, Waltham, MA, USA, or Lonza, Basel, Switzerland) supplemented with 5% (v/v) Human AB serum (Sigma-Aldrich, St. Louis, MO, USA), 1% (v/v) Penicillin/Streptomycin, 1 mM sodium pyruvate (Lonza), 1% (v/v) non-essential amino acids (Cegrogen Biotech, Stadtallendorf, Germany), 15 mM HEPES buffer (Sigma-Aldrich), and 55 μM 2-mercaptoethanol (AppliChem, Darmstadt, Germany).

### *In vitro* expansion of human Vδ2$^+$ T cells

Human Vδ2$^+$ T cells were expanded from PBMCs of healthy donors similar to published protocols [41–43]. Briefly, freshly isolated PBMCs (1 x 10$^6$ cell/ml, 5 ml/well) were cultured with 5 μM Zoledronic acid (Zometa, Novartis) in the presence of 100 IU/ml human recombinant IL-2 (Proleukin, Novartis) for 13 days. IL-2 was replenished every other day and from day 6 onwards the concentration was increased to 200 IU/ml. The cultures were performed in RPMI1640 (Gibco or Lonza) supplemented with 5% (v/v) Human AB serum (Sigma-Aldrich), 1% (v/v) Penicillin/Streptomycin (Gibco), 1 mM sodium pyruvate (Lonza), 1% (v/v) non-essential amino acids (Cegrogen), 15 mM HEPES buffer (Sigma-Aldrich), and 55 μM 2-mercaptoethanol (AppliChem).

### Human macrophage generation

Purified monocytes were cultured in RPMI1640 (Gibco or Lonza) medium containing 5% FCS (Corning, NY, USA), 1% Penicillin/Streptomycin (Gibco), and 10 ng/ml human recombinant M-CSF (Peprotech, London, UK) for their differentiation into macrophages. 3 x 10$^6$ cells/well were seeded in low-attachment 6-well plates (Corning) and incubated for 7 days at 5% CO$_2$ and 37°C. Macrophages were collected and cultured in 24-well cell culture plates as 5 x 10$^5$ macrophages/well for 24 hours in RPMI1640 medium containing 5% FCS, 1% Penicillin/Streptomycin. On the following day, the cell culture media was replaced with fresh media and macrophages were stimulated with the relevant polarization factors as described below. The cells were verified to be over 90% CD68$^+$ by flow cytometry.

### *In vitro* stimulation

Splenocytes were stimulated *in vitro* with 50 ng/ml PMA and 1 μg/ml ionomycin (both Sigma-Aldrich) for 4 h at 37°C in the presence of both Brefeldin A (GolgiPlug) and Monensin (Golgi-Stop, both BD Biosciences). As GolgiPlug and GolgiStop were used together, half the amount recommended by the manufacturer where used, as suggested previously [44]. Cells were stimulated in complete$^{mouse}$ RPMI medium (RPMI 1640 supplemented with 10% (v/v) FCS (Corning), 1% (v/v) Pen-Strep-Glutamine (10.000 U/ml penicillin, 10.000 μg/ml streptomycin, 29.2 mg/ml L-glutamine (Gibco)) and 50 μM 2-mercaptoethanol (AppliChem), containing 0.42

mM $Ca^{2+}$). For human studies, freshly isolated PBMCs or expanded cell populations were stimulated *in vitro* with 25 ng/ml PMA and 1 μg/ml PMA for 4 h at 37˚C in the presence of Brefeldin A or monensin. Cells were stimulated in complete[human] RPMI medium (RPMI 1640 supplemented with 10% (v/v) FCS, 1% (v/v) Penicillin/Streptomycin, 1 mM sodium pyruvate (Lonza), 1% (v/v) non-essential amino acids (Cegrogen), 15 mM HEPES buffer (Sigma-Aldrich), and 55 μM 2-mercaptoethanol (AppliChem). Considering the $Ca^{2+}$ found in the FCS (3.9 mM) [21] the supplemented RPMI medium contained approx. 0.8 mM $Ca^{2+}$ (RPMI[norm]). To obtain the RPMI[suppl] medium containing elevated $Ca^{2+}$ concentrations, 1 mM $CaCl_2$ (Sigma-Aldrich) was added to the RPMI[norm] medium (i.e. approx. 1.8 mM $Ca^{2+}$).

## Human macrophage polarization

Macrophages were stimulated (i) for M1 polarization with 100 ng/ml LPS (InvivoGen, San Diego, CA, USA) and 20 ng/ml IFNγ (R&D Systems) for 6h, 12h, or 24h; (ii) for M2a with 20 ng/ml IL-4 (R&D Systems) for 24h or 48h; or (iii) for M2c with 20 ng/ml IL-10 (R&D Systems) for 24h or 48h. The particular incubation times for the experiments are specified in the figures.

## Statistical analysis

Results are expressed as mean ± standard error of the mean (SEM). Normal distribution was tested using D'Agastino–Pearson Test. Statistical comparisons were drawn using either a two-tailed paired Student's t-test (normally distributed data) or Wilcoxon matched-pairs signed rank test (not normally distributed data) (Excel, Microsoft Corporation; GraphPad Prism, GraphPad Software). p-values <0.05 were considered significant and are indicated with ($^*$p < 0.05, $^{**}$p < 0.01, $^{***}$p < 0.001, $^{****}$p <0.0001. Each experiment was repeated at least twice, and background values were subtracted. Graphs were generated with GraphPad Prism (GraphPad Software).

## Supporting information

**S1 Fig. Gating strategy to identify leukocytes and macrophage purity. (A, B)** Exemplary dot plots illustrating the gating strategy employed to identify the indicted (A) mouse or (B) human lymphoid cells. **(C)** Purity of the differentiated macrophages: human PBMC-derived monocytes were incubated for 7 days in RPMI1640 medium containing 10 ng/ml M-CSF for macrophage differentiation and the percentage of CD68$^+$ macrophages was determined by intracellular staining. Left: representative flow cytometry data (blue = undifferentiated; red = differentiated); Right: Summary data (n = 3; un-diff = undifferentiated; diff = differentiated). **(D-F)** Evaluation of macrophage polarization. Primary human monocyte-derived macrophages were left unstimulated (UT) or stimulated for 12 h (D) with 100 ng/ mL LPS and 20 ng/mL IFN-γ for M1 polarization (M1), (E) with 20 ng/mL IL-4 for M2a polarization (M2a), or (F) with 20 ng/ mL IL-10 for M2c polarization (M2c). Expression of indicated surface markers was analyzed by flow cytometry. The bar graphs indicate mean fluorescent intensity (MFI). The biological replicates of 6 independent donors pooled from 2 independent experiments are shown.
(PDF)

**S2 Fig. Stimulation of murine *i*NKT cells, γδ T cells, or B cells in RPMI$^{suppl}$ has no negative impact on cell viability.** Splenocytes from C57BL/6 mice were stimulated 4 h with 50 ng/ml PMA and 1 μg/ml ionomycin in either normal RPMI1640 medium (RPMI[norm]) or RPMI1640 medium supplemented with 1 mM $Ca^{2+}$ (RPMI[suppl]). **(a)** *i*NKT cells; **(b)** Vδ2$^+$ T cells; and **(c)** B cells were stained and analysed by flow cytometry. The bar graphs show the relative

percentages of cells positive for LIVE/DEAD Fixable Blue Dead Cell Stain, indicating dead cells. Data were pooled from three independent experiments with three mice per group per experiment (n = 9).
(TIF)

**S3 Fig. Ca²⁺ supplementation *in vitro* has no impact on the production of some cytokines by mouse γδ T and B cells. (a-c)** Splenocytes from C57BL/6 mice were stimulated 4 h with 50 ng/ml PMA and 1 μg/ml ionomycin in either normal RPMI1640 medium (RPMI$^{norm}$) or RPMI1640 medium supplemented with 1 mM Ca²⁺ (RPMI$^{suppl}$). The production of **(a)** IL-10 and **(b)** IL-17A by γδ T cells (live CD19/CD45R⁻ CD4⁻ CD8α⁻ CD3ε⁺ γδTCR⁺ cells) and the production of **(c)** IFNγ by B cells (live CD3ε⁻ CD4⁻ CD8α⁻ CD19/CD45R⁺ cells) was analysed by ICCS. Summary graphs (left panels) and representative data (right panels) from gated γδ T cells and B cells are shown, respectively. Data were pooled from three independent experiments with three mice per group per experiment (n = 9).
(TIF)

**S4 Fig. Ca²⁺ supplementation *in vitro* has no impact on the expression of CD69 and the production IL-2, IFNγ, IL-4, and IL-17 by primary human *i*NKT cells.** PBMCs were isolated from the residual leukocyte units of healthy donors. PBMCs were stimulated for 4 h with 25 ng/ml PMA and 1 μg/ml ionomycin in either normal RPMI1640 medium (RPMI$^{norm}$) or RPMI1640 medium supplemented with 1 mM Ca²⁺ (RPMI$^{suppl}$). Human Vα24*i* NKT cells (live CD14⁻ CD20⁻ CD3⁺ 6B11⁺ cells) were analysed for the expression of the activation marker **(a)** CD69 and the production of the cytokines **(b)** IL-2, **(c)** IFNγ, **(d)** IL-4, and **(e)** IL-17. Summary graphs (left panels) and representative data (right panels) from gated *i*NKT cells are shown, respectively. Data were pooled from three independent experiments with three samples each (n = 9).
(JPG)

**S5 Fig. Ca²⁺ supplementation *in vitro* has no impact on expression of CD69 on human primary Vγ2⁺ T cells, MAIT cells, and B cells, and on the production of the IL-2 by human monocytes and IL-4 by Vγ2⁺ T cells.** PBMCs were isolated from residual leukocyte units of healthy donors and were stimulated for 4 h with 25 ng/ml PMA and 1 μg/ml ionomycin in either normal RPMI1640 medium (RPMI$^{norm}$) or RPMI1640 medium supplemented with 1 mM Ca²⁺ (RPMI$^{suppl}$). Human Vδ2⁺ T cells (live CD14⁻ CD20⁻ CD3⁺ γδTCR$^{low}$ or Vγ2⁺ cells) were analysed for the expression of **(a)** CD69 and the production of **(b)** IL-2 and **(c)** IL-4. **(d)** Human MAIT cells (live CD14⁻ CD20⁻ CD3⁺ Vα7.2⁺ CD161⁺ cells) were analysed for the expression of CD69. **(e)** Human B cells (CD14⁻ CD3⁻ CD20⁺ cells) were analysed for the expression of CD69. **(f)** Human monocytes (CD3⁻ CD20⁻ CD14⁺ cells) were analysed for the production of IL-2. Summary graphs (left panels) and representative data (right panels) are shown for the cytokine data. Data were pooled from three (Vδ2⁺ T cells; n = 9) and four (MAIT cells, B cells, monocytes; n = 12) independent experiments with three samples each.
(JPG)

**S6 Fig. Ca²⁺ supplementation *in vitro* has no impact on the expression of CD69 by expanded Vγ2⁺ T cells and on the production of some cytokines by expanded human *i*NKT and Vγ2⁺ T cells.** *i*NKT cells were expanded *ex vivo* in the presence of αGalCer. Vγ2⁺ T cells were expanded *in vitro* in the presence of Zoledronic acid. The expanded cells were stimulated for 4 h with 25 ng/ml PMA and 1 μg/ml ionomycin in either normal RPMI1640 medium (RPMI$^{norm}$) or RPMI1640 medium supplemented with 1 mM Ca²⁺ (RPMI$^{suppl}$). Vα24*i* NKT cells (live CD14⁻ CD20⁻ CD3⁺ 6B11⁺ cells) were analysed for the production of (a) IL-2, (b) IFNγ, (c) IL-4, and (d) IL-17 were measured by ICCS. Human Vδ2⁺ T cells (live CD14⁻ CD20⁻

CD3$^+$ Vγ2$^+$ cells) were analysed for the expression of (e) CD69 and the production of (f) TNF, (g) IL-4. Summary graphs (left panels) and representative data (right panels) from gated *i*NKT cells and Vγ2$^+$ T cells are shown, respectively. Data were pooled from four and three independent experiments with three samples each for *i*NKT cells (n = 12) and Vγ2$^+$ T cells (n = 9), respectively.
(JPG)

**S7 Fig. Stimulation of human lymphoid cells in RPMI$^{suppl}$ has no negative effect on cell viability.** PBMCs were isolated from residual leukocyte units of healthy donors and were stimulated either directly (a-d) or after in vitro expansion of indicated cells (e, f). The cells were stimulated for 4 h with 25 ng/ml PMA and 1 μg/ml ionomycin in either normal RPMI1640 medium (RPMI$^{norm}$) or RPMI1640 medium supplemented with 1 mM Ca$^{2+}$ (RPMI$^{suppl}$). Primary (a) *i*NKT cells, (b) Vδ2$^+$ T cells, (c) B cells; and (d) MAIT cells, (e) monocytes, or *in vitro* expanded (f) *i*NKT cells and (g) Vδ2$^+$ T cells were stained and analysed by flow cytometry. The bar graphs show the relative percentages of cells positive for LIVE/DEAD Fixable Blue Dead Cell Stain, indicating dead cells. Data were pooled from three (a-b, g) or four (c—f) independent experiment with three samples each (n = 9–12).
(JPG)

**S8 Fig. Ca$^{2+}$ supplementation *in vitro* increases cell death of M0 macrophages.** Primary human monocyte-derived macrophages were cultured for 6, 12, 24, or 48 hours in either normal RPMI1640 medium (RPMI$^{norm}$) or RPMI1640 medium supplemented with 1 mM Ca$^{2+}$ (RPMI$^{suppl}$). The bar graphs show the relative percentages of cells positive for LIVE/DEAD Fixable Blue Dead Cell Stain (LDB$^+$ cells), indicating dead cells. Data shown are means ± SEM of biological replicates of six donors, pooled from two independent experiments with similar results.
(JPG)

**S9 Fig. Ca$^{2+}$ supplementation *in vitro* affects the expression of macrophage polarization markers.** Primary human monocyte-derived macrophages were cultured untreated (MØ) or polarized into M1 (100ng/ml LPS, 20 ng/ml IFNγ, 12 hours), M2a (20 ng/ml IL-4, 24 hours), or M2c (20ng/ml IL-10, 24 hours) macrophages in either normal RPMI1640 medium (RPMI$^{norm}$) or RPMI1640 medium supplemented with 1 mM Ca$^{2+}$ (RPMI$^{suppl}$). The surface expression (mean fluorescent intensity, MFI) of indicated **(a)** M1 markers (HLA-DR, CD86, CD64), **(b)** M2a markers (CD200R, CD206), and an **(c)** M2c marker (CD163) are shown. The data shown are means ± SEM of biological replicates of six donors, pooled from two independent experiments with similar results.
(JPG)

**S1 Table. Details on the antibodies used in this study.**
(PDF)

## Acknowledgments

The authors wish to thank the Flow Cytometry Core Facility and the vivarium at the Izmir Biomedicine and Genome Center (IBG) for excellent technical assistance. We are grateful to the NIH Tetramer Core Facility (Emory University, Atlanta, USA) for providing the mouse CD1d/PBS57 tetramers.

## Author Contributions

**Conceptualization:** Yusuf Cem Eskiocak, Gerhard Wingender.

**Formal analysis:** Yusuf Cem Eskiocak, Zeynep Ozge Ayyildiz, Sinem Gunalp, Asli Korkmaz, Duygu Sag, Gerhard Wingender.

**Funding acquisition:** Duygu Sag, Gerhard Wingender.

**Investigation:** Yusuf Cem Eskiocak, Zeynep Ozge Ayyildiz, Sinem Gunalp, Asli Korkmaz, Derya Goksu Helvaci, Gerhard Wingender.

**Methodology:** Yusuf Cem Eskiocak, Zeynep Ozge Ayyildiz, Sinem Gunalp, Asli Korkmaz, Duygu Sag, Gerhard Wingender.

**Project administration:** Gerhard Wingender.

**Resources:** Yavuz Dogan.

**Supervision:** Duygu Sag, Gerhard Wingender.

**Validation:** Yusuf Cem Eskiocak, Zeynep Ozge Ayyildiz, Gerhard Wingender.

**Visualization:** Yusuf Cem Eskiocak, Zeynep Ozge Ayyildiz, Sinem Gunalp, Asli Korkmaz, Gerhard Wingender.

**Writing – original draft:** Yusuf Cem Eskiocak, Zeynep Ozge Ayyildiz, Gerhard Wingender.

**Writing – review & editing:** Yusuf Cem Eskiocak, Zeynep Ozge Ayyildiz, Duygu Sag, Gerhard Wingender.

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
