## [Decision Letter · Decision Letter 0]

28 Sep 2022

PONE-D-22-15126The Ca2+ concentration in vitro impacts the cytokine production of mouse and human lymphoid cells and the polarization of human macrophagesPLOS ONE

Dear Dr. Wingender,

Thank you for submitting your manuscript to PLOS ONE. After careful consideration, we feel that it has merit but does not fully meet PLOS ONE’s publication criteria as it currently stands. Therefore, we invite you to submit a revised version of the manuscript that addresses the points raised during the review process.

We look forward to receiving your revised manuscript.

Kind regards,

Nazmul Haque

Academic Editor

PLOS ONE

Journal Requirements:

2. To comply with PLOS ONE submissions requirements, in your Methods section, please provide additional information on the animal research and ensure you have included details on (1) methods of sacrifice, (2) methods of anesthesia and/or analgesia, and (3) efforts to alleviate suffering

Reviewers' comments:

Reviewer's Responses to Questions

**Comments to the Author**

1. Is the manuscript technically sound, and do the data support the conclusions?

Reviewer #1: No

2. Has the statistical analysis been performed appropriately and rigorously? 

Reviewer #1: Yes

3. Have the authors made all data underlying the findings in their manuscript fully available?

Reviewer #1: Yes

4. Is the manuscript presented in an intelligible fashion and written in standard English?

Reviewer #1: Yes

5. Review Comments to the Author

Reviewer #1: The current manuscript is well written. However, the following points need to be addressed before publishing -

1# If the author shift the word ‘in vitro’ to the end of the sentence the title sounds more familiar.

2# In the method section author wrote that ‘Antibodies were purchased 357 from either BioLegend (USA), BD Biosciences (USA), or eBiosciences (USA).’ However, there is an international format for writing chemical reagents in a scientific paper. The author needs to be specific with company name by particular reagent. Furthermore, the author is suggested to recheck all manuscript to follow the same rule while mention company manes.

3# The ‘Cell preparation’ section in Methods seeking brief explanation of procedures of cell isolations.

4# The author used 1 mM CaCl2 for stimulating cells in vitro. What is the rationale of using this single concentration? Is this stimulation is dose independent?

5# Under the section ‘Human macrophage polarization’, the author wrote that ‘The incubation times are specified in the text.’ However, they are suggested to write it here. There is no problem if they want to keep it in the text as an addition.

6# The authors are strongly suggested to add positive and negative controls in their experiment.

6. PLOS authors have the option to publish the peer review history of their article (what does this mean?). If published, this will include your full peer review and any attached files.

Reviewer #1: No

---

## [Author Response · Author response to Decision Letter 0]

11 Oct 2022

Editors

Response: We followed PLOS ONE’s guidelines to our best knowledge throughout.

2. To comply with PLOS ONE submissions requirements, in your Methods section, please provide additional information on the animal research and ensure you have included details on (1) methods of sacrifice, (2) methods of anesthesia and/or analgesia, and (3) efforts to alleviate suffering.

Response: We added the method of sacrifice of the mice (cervical dislocation) to the Material and Methods section. All mouse experiments were performed ex vivo, therefore, no anaesthesia was required. However, we added a statement that we followed 3R guidelines to the relevant section in the Material and Methods section.

Response: We ticked the option “URLs/accession numbers/DOIs” during the online submission by mistake. No data related to this manuscript need to be deposited and, therefore, we removed the tick now during the resubmission.

4. We note that the grant information you provided in the ‘Funding Information’ and ‘Financial Disclosure’ sections do not match. When you resubmit, please ensure that you provide the correct grant numbers for the awards you received for your study in the ‘Funding Information’ section.

Response: Unfortunately, this point is not entirely clear to us. We found on the editorialmanager.com website, the ‘Funding Information’ section (tab ‘Manuscript Data’) but not the ‘Financial Disclosure’ section. All information provided in the ‘Funding Information’ section is complete and correct. No information on funding was initially added to the manuscript text as the PLOS ONE guidelines (https://journals.plos.org/plosone/s/submission-guidelines) explicitly state “Do not include funding sources in the Acknowledgments or anywhere else in the manuscript file. Funding information should only be entered in the financial disclosure section of the submission system”. Upon inquiry with PLOS ONE (plosone@plos.org), we were informed that “The Financial Disclosure written in your manuscript must match your Funding Information in the Manuscript Data Tab. Therefore, we encourage you to correct your Financial Disclosure in your manuscript.” Following this advice, we now added a ‘Financial disclosure’ section to the main text.

Reviewer #1 

1. If the author shift the word ‘in vitro’ to the end of the sentence the title sounds more familiar.

Response: We thank the reviewer for this suggestion and the title was changed accordingly.

2. In the method section author wrote that ‘Antibodies were purchased 357 from either BioLegend (USA), BD Biosciences (USA), or eBiosciences (USA).’ However, there is an international format for writing chemical reagents in a scientific paper. The author needs to be specific with company name by particular reagent. Furthermore, the author is suggested to recheck all manuscript to follow the same rule while mention company manes.

Response: We thank the reviewer to bring this ambiguity to our attention. We double-checked that Material & Method section and belief to have stated now the specific vendor for every reagent mentioned, including for the utilized antibodies.

3. The ‘Cell preparation’ section in Methods seeking brief explanation of procedures of cell isolations.

Response: We extended the information provided under ‘Cell Preparation’ to make the paragraph self-explanatory without the need to revert to previous publications. 

4. The author used 1 mM CaCl2 for stimulating cells in vitro. What is the rationale of using this single concentration? Is this stimulation is dose independent?

Response: It was shown previously (Zimmermann et al. 2015, PMID: 25545753) that supplementing RPMI media with 1 mM of CaCl2 is required to obtain the maximal cytokine production by stimulated conventional CD4+ T cells. The goal of our study was to clarify whether this observation would also be relevant for unconventional T cells (iNKT cells, MAIT cells, �� T cells) and human macrophages (as an example of myeloid cells). Therefore, we think that Ca2+-titration experiments would not be in line with our aim and would distract from our main finding.

5. Under the section ‘Human macrophage polarization’, the author wrote that ‘The incubation times are specified in the text.’ However, they are suggested to write it here. There is no problem if they want to keep it in the text as an addition.

Response: We thank the reviewer for bringing this omission to our attention. We added now the range of incubation times to the Material & Method section.

6. The authors are strongly suggested to add positive and negative controls in their experiment.

Response: We agree with the reviewer that negative and positive controls are essential for all experiments. All our experiments included an untreated (e.g. unstimulated) control (i.e. negative control). Our positive control was the cell stimulation with PMA/ionomycin (lymphoid cells, monocytes), LPS/IFN� (M1), IL-4 (M2a), or IL-10 (M2c) in standard RPMI medium. Our experimental groups were the stimulation with Ca2+-supplemented RPMI medium. Therefore, all our experiments were performed with the relevant negative and positive controls.

---

## [Decision Letter · Decision Letter 1]

28 Nov 2022

PONE-D-22-15126R1The Ca2+ concentration impacts the cytokine production of mouse and human lymphoid cells and the polarization of human macrophages in vitroPLOS ONE

Dear Dr. Wingender,

Thank you for submitting your manuscript to PLOS ONE. After careful consideration, we feel that it has merit but does not fully meet PLOS ONE’s publication criteria as it currently stands. Therefore, we invite you to submit a revised version of the manuscript that addresses the points raised during the review process.

We look forward to receiving your revised manuscript.

Kind regards,

Nazmul Haque

Academic Editor

PLOS ONE

Reviewers' comments:

Reviewer's Responses to Questions

**Comments to the Author**

1. If the authors have adequately addressed your comments raised in a previous round of review and you feel that this manuscript is now acceptable for publication, you may indicate that here to bypass the “Comments to the Author” section, enter your conflict of interest statement in the “Confidential to Editor” section, and submit your "Accept" recommendation.

Reviewer #1: (No Response)

Reviewer #2: (No Response)

2. Is the manuscript technically sound, and do the data support the conclusions?

Reviewer #1: (No Response)

Reviewer #2: Yes

3. Has the statistical analysis been performed appropriately and rigorously? 

Reviewer #1: (No Response)

Reviewer #2: Yes

4. Have the authors made all data underlying the findings in their manuscript fully available?

Reviewer #1: (No Response)

Reviewer #2: No

5. Is the manuscript presented in an intelligible fashion and written in standard English?

Reviewer #1: (No Response)

Reviewer #2: No

6. Review Comments to the Author

Reviewer #1: The author answered that 'It was shown previously (Zimmermann et al. 2015, PMID: 25545753) that supplementing RPMI media with 1 mM of CaCl2 is required to obtain the maximal cytokine production by stimulated conventional CD4+ T cells. '

They are suggested to add this explanation in their manuscript in the most suitable place.

Reviewer #2: The manuscript entitled ‘The Ca2+ concentration in vitro impacts the cytokine production of mouse and human lymphoid cells and the polarization of human macrophages’, by Yusuf Cem Eskiocak et al gives evidence for an impact of Ca 2+ concentration in culture media on the functional response of immune cells in ex vivo assay systems.

The manuscript highlights an observation known for decades but not addressed in detail.

Major comments:

1.) The authors use standard isolation procedures to isolate T cell subsets and they polarize monocytes to macrophages M1/M2 but do not show the success of their manipulation. Pheno/genotypic markers need to be used to precisely characterize the invariant NK-T cells, mucosal-associated invariant T cells, γδ T cells after isolation/stimulation. The same is true for monocyte-macrophage differentiation.

2.) I have a lot of issues open concerning the description of the methods and materials used.

a. For human samples an approval number of the Ethic committee is provided but this is not the case for animals.

b. ARRIVE guidelines are cited with no reference

c. All antibodies used were given with a clone identification but without the given concentration and conjugation with either Brilliant Ultra Violet 395, Pacific Blue, Violet 500, Brilliant Violet 570,358 Brilliant Violet 605, Brilliant Violet 650, Brilliant Violet 711, Brilliant Violet 785, FITC, PerCP-359 Cy5.5, PerCP-eF710, PE, PE-CF594, PE-Dazzle594, PE-Cy7, APC, AF647, eF660, AF700, 360 APC-Cy7, or APC-eF780 and therefor it is hard to follow.

d. No concentration is given on the blocking mAbs.

e. What kind of Flow Cytometer is used and which program for calculation and graphical display?

f. When it comes to statistic paired tests are not appropriate and it is easy for the authors to provide unpaired alternative tests.

3.) The reviewer has the impression that the authors show too many Figures 1-7 and additional half-a-dozen Figures in the Supplements. I would recommend to extract the important issues and combine Figures, describe the irrelevant in the results and add the necessary controls of T cell and macrophage subsets in the Supplements.

Minor comments:

1.) When providing information on materials used, the authors should provide more information. BioLegend should read BioLegend® Inc., San Diego, CA for the first time and thereafter BioLegend every second/third time. I ‘know’ this is work.

2.) Also the English language needs to be improved for clarity.

7. PLOS authors have the option to publish the peer review history of their article (what does this mean?). If published, this will include your full peer review and any attached files.

Reviewer #1: No

Reviewer #2: No

---

## [Author Response · Author response to Decision Letter 1]

29 Dec 2022

Reviewer #1 

The author answered that 'It was shown previously (Zimmermann et al. 2015, PMID: 25545753) that supplementing RPMI media with 1 mM of CaCl2 is required to obtain the maximal cytokine production by stimulated conventional CD4+ T cells.' They are suggested to add this explanation in their manuscript in the most suitable place.

Response: We thank the reviewer for this suggestion and we highlighted this point better now in the introduction. 

Reviewer #2

The manuscript highlights an observation known for decades but not addressed in detail.

Response: That calcium in the medium is require for in vitro responses by immune cells is indeed know for decades, but this is not the topic of our study. That the calcium concentration in the commonly used RPMI1640 medium is insufficient for maximal cytokine responses of conventional T cells was first reported in 2015. We now extent this finding to other, unconventional T cells (iNKT cells, MAIT cells, �� T cells) and to macrophages as an example of myeloid cells. We do believe that our data are of significant interest to researchers utilizing in vitro experiments to study unconventional T cells and macrophages.

1.) The authors use standard isolation procedures to isolate T cell subsets and they polarize monocytes to macrophages M1/M2 but do not show the success of their manipulation. Pheno/genotypic markers need to be used to precisely characterize the invariant NK-T cells, mucosal-associated invariant T cells, γδ T cells after isolation/stimulation. The same is true for monocyte-macrophage differentiation.

Response: To illustrate our gating and the purity of the analysed cell populations, we now included a new supplemental figure (S1 Fig).

2.) I have a lot of issues open concerning the description of the methods and materials used.

a. For human samples an approval number of the Ethic committee is provided but this is not the case for animals.

Response: This information was already provided in the manuscript, under the section ’Mice’: “All mouse experiments were performed with prior approval by the institutional ethic committee (‘Ethical Committee on Animal Experimentation’ of the Izmir Biomedicine and Genome Center, approval number: 19/2016) in accordance with national laws and policies.”

b. ARRIVE guidelines are cited with no reference

Response: We thank the reviewer for pointing out this inadvertent omission. We added the reference in the revised version of the manuscript.

c. All antibodies used were given with a clone identification but without the given concentration and conjugation with either Brilliant Ultra Violet 395, Pacific Blue, Violet 500, Brilliant Violet 570,358 Brilliant Violet 605, Brilliant Violet 650, Brilliant Violet 711, Brilliant Violet 785, FITC, PerCP-359 Cy5.5, PerCP-eF710, PE, PE-CF594, PE-Dazzle594, PECy7, APC, AF647, eF660, AF700, 360 APC-Cy7, or APC-eF780 and therefor it is hard to follow.

Response: To further clarify this point, we added a new supplemental table showing the clone, conjugate, source, ID, and dilution for each antibody used in this study.

d. No concentration is given on the blocking mAbs.

Response: We assume the reviewer refers to the reagents used to block the Fc receptors. We thank the reviewer for pointing out this inadvertent omission. We added now the clarification “… according to the manufacturers’ recommendations”

e. What kind of Flow Cytometer is used and which program for calculation and graphical display?

Response: We thank the reviewer for pointing out this inadvertent omission. This information has been added now to the Material and Methods section.

f. When it comes to statistic paired tests are not appropriate and it is easy for the authors to provide unpaired alternative tests.

Response: The use of paired tests for the statistical analysis of the reported data (cell frequencies in percentages, geometric MFIs) is in line with the common procedure in the field. Furthermore, we previously verified internally that such data are normally distributed (D’Agostino - Pearson omnibus normality test after combining values from repetitive experiments (n>10, GraphPad Prism)), demonstrating in our opinion that a paired test is appropriate. However, should the reviewer outline for which experiment the usage of a paired test is inappropriate and why, we are happy to revaluate our analysis. 

3.) The reviewer has the impression that the authors show too many Figures 1-7 and additional half-a-dozen Figures in the Supplements. I would recommend to extract the important issues and combine Figures, describe the irrelevant in the results and add the necessary controls of T cell and macrophage subsets in the Supplements.

Response: We understand the concern of the reviewer; however, we think that combining figures or removing parts of them would reduce the clarity or required information content, respectively. 

Minor comments:

1.) When providing information on materials used, the authors should provide more information. BioLegend should read BioLegend® Inc., San Diego, CA for the first time and thereafter BioLegend every second/third time. I ‘know’ this is work.

Response: We did not see the usage of ® in other recent PLOS One publications and, similar, the location of the company headquarters appears not to be a journal requirement. However, we included now the requested information on the company headquarters in the revised manuscript.

2.) Also the English language needs to be improved for clarity.

Response: If the reviewer refers to the English in which the manuscript is written, then we are surprised to see this comment, as we are confident that the understandability and clarity of the English utilized throughout meets PLOS One standards, not alone as three papers from the laboratory were recently (2021, 2022) published in PLOS One. However, should the reviewer point out specific issues, we would be happy to improve the text were necessary.

---

## [Decision Letter · Decision Letter 2]

7 Feb 2023

The Ca2+ concentration impacts the cytokine production of mouse and human lymphoid cells and the polarization of human macrophages in vitro

PONE-D-22-15126R2

Dear Dr. Wingender,

We’re pleased to inform you that your manuscript has been judged scientifically suitable for publication and will be formally accepted for publication once it meets all outstanding technical requirements.

Kind regards,

Nazmul Haque

Academic Editor

PLOS ONE

Additional Editor Comments (optional):

Reviewers' comments:

Reviewer's Responses to Questions

**Comments to the Author**

1. If the authors have adequately addressed your comments raised in a previous round of review and you feel that this manuscript is now acceptable for publication, you may indicate that here to bypass the “Comments to the Author” section, enter your conflict of interest statement in the “Confidential to Editor” section, and submit your "Accept" recommendation.

Reviewer #1: (No Response)

Reviewer #2: All comments have been addressed

2. Is the manuscript technically sound, and do the data support the conclusions?

Reviewer #1: (No Response)

Reviewer #2: Yes

3. Has the statistical analysis been performed appropriately and rigorously? 

Reviewer #1: (No Response)

Reviewer #2: Yes

4. Have the authors made all data underlying the findings in their manuscript fully available?

Reviewer #1: (No Response)

Reviewer #2: Yes

5. Is the manuscript presented in an intelligible fashion and written in standard English?

Reviewer #1: (No Response)

Reviewer #2: Yes

6. Review Comments to the Author

Reviewer #1: (No Response)

Reviewer #2: Thank you for addressing the majority of the issues addressed. Also the modifications are well done and adequate.

7. PLOS authors have the option to publish the peer review history of their article (what does this mean?). If published, this will include your full peer review and any attached files.

Reviewer #1: No

Reviewer #2: **Yes: **Fischer Michael B.

---

## [Editor Report · Acceptance letter]

15 Feb 2023

PONE-D-22-15126R2 

The Ca^2+^ concentration impacts the cytokine production of mouse and human lymphoid cells and the polarization of human macrophages *in vitro*

Dear Dr. Wingender:

I'm pleased to inform you that your manuscript has been deemed suitable for publication in PLOS ONE. Congratulations! Your manuscript is now with our production department. 

Kind regards, 

on behalf of

Dr. Nazmul Haque 

Academic Editor

PLOS ONE